# Platelet Activity and Its Correlation with Inflammation and Cell Count Readings in Chronic Heart Failure Patients with Reduced Ejection Fraction

**DOI:** 10.3390/medicina57020176

**Published:** 2021-02-18

**Authors:** Aušra Mongirdienė, Jolanta Laukaitienė, Vilius Skipskis, Lolita Kuršvietienė, Julius Liobikas

**Affiliations:** 1Department of Biochemistry, Medicine Academy, Lithuanian University of Health Sciences, Eiveniu Str. 4, LT-50103 Kaunas, Lithuania; jolanta_laukaitiene@yahoo.co.uk (J.L.); lolita.kursvietiene@lsmuni.lt (L.K.); julius.liobikas@lsmuni.lt (J.L.); 2Cardiology Clinic, University Hospital, Lithuanian University of Health Sciences, Eiveniu Str. 2, LT-50161 Kaunas, Lithuania; 3Laboratory of Molecular Cardiology, Institute of Cardiology, Lithuanian University of Health Sciences, Eiveniu Str. 4, LT-50103 Kaunas, Lithuania; skipskis@gmail.com; 4Laboratory of Biochemistry, Neuroscience Institute, Lithuanian University of Health Sciences, LT-50161 Kaunas, Lithuania

**Keywords:** platelet, mean platelet volume, platelet aggregation, heart failure, monocyte, neutrophil, lymphocyte

## Abstract

*Background and objectives*: There has been an increasing interest in the role of inflammation in thrombosis complications in chronic heart failure (HF) patients. The incidence of thrombosis in HF has been shown to be the highest in patients classified as NYHA IV (New York Heart association). It is stated that inflammation is regulated by platelet-induced activation of blood leukocytes. We aimed to compare the platelet and cell count readings in chronic HF with reduced ejection fraction (HFrEF) patients according to NYHA functional class and to evaluate the correlation between those readings. *Materials and methods*: A total of 185 patients were examined. The results of heart echoscopy (TEE) testing; fibrinogen, N-terminal pro b-type natriuretic peptide (NT-proBNP), C reactive protein (CRP), and cortisol concentrations; complete blood counts; and a 6 min walking test were assessed and platelet aggregation was determined. *Results*: Mean platelet volume (MPV) increased with deterioration of a patient’s state (*p* < 0.005). Lymphocyte count and percentage were the lowest in the NYHA IV group (*p* < 0.005). Neutrophil and monocyte percentage and count were the highest (*p* < 0.045) in the NYHA IV group. Adenosine diphosphate (ADP)- and ADR-induced platelet aggregation was higher in the NYHA III group compared to NYHA II and I groups (*p* < 0.023). NYHA functional class correlated with mean platelet volume (MPV) (r = 0.311, *p* = 0.0001), lymphocyte count (r = −0.186, *p* = 0.026), monocyte count (*p* = 0.172, *p* = 0.041), and percentage (r = 0.212, *p* = 0.011). CRP concentration correlated with NT-proBNP (r = 0.203, *p* = 0.005). MPV correlated with fibrinogen concentration (r = 0.244, *p* = 0.004). *Conclusions*: (1) MPV could be considered as an additional reading reflecting a patient’s condition, however the use of MPV to identify patients at risk of hypercoagulable state should be evaluated in more extensive studies; (2) increased neutrophil and monocyte counts could indicate a higher inflammatory state in chronic HFrEF.

## 1. Introduction

Reduced heart function as a result of chronic heart failure (CHF) has recently been determined to be an independent risk factor for venous thromboembolism [1]. The incidence of deep venous thrombosis (DVT) in heart failure patients has been shown to be as high as 11.2%, with the incidence being the highest in patients classified as NYHA IV (New York Heart association) IV (NYHA II: 4.4%, NYHA III: 4.8%, NYHA IV: 25.5%, *p* < 0.01) [2]. Venous thrombosis (VTE) in patients with heart failure varies from no risk to high risk (2.9–32.4%) [3]. VTE risk correlates with the severity of heart failure [4]. Increased platelet activation and thrombin formation in patients with CHF may lead to a hypercoagulable state and contribute to thrombogenesis [5]. Despite the abundance of reasons known to lead to thrombosis, the exact pathophysiological mechanisms of thrombosis in CHF remain to be discovered.

There has been a growing interest in the role of inflammation in thrombosis complications in CHF. The platelet’s involvement in the interplay between hemostasis, thrombosis and inflammation is important in disease occurrence [6]. Chronic inflammation is more common in HF patients [7]. Inflammatory mediators (TNF-α, IL-6, CD-154, C-reactive protein (CRP)) have been described as being elevated in CHF [8,9].

It has been stated that inflammation processes are regulated by platelet-induced activation of blood leukocytes (monocytes, lymphocytes, neutrophils, basophils, and eosinophils) [10]. A mean platelet volume (MPV) is presented as a reading, which can be used for prognosis of patients in various clinical situations [11]. MPV values in HF patients has been found to be higher than in age–sex-matched controls [12].

While myocardial damage in heart failure patients with reduced ejection fraction (HFrEF) has been shown to be driven by oxidative stress, inflammation is a recognized factor in disease progression in both HFrEF and HFpEF (heart failure with preserved ejection fraction) [13]. However, to the best of our knowledge, there are no data in the literature about differences in cell count, which take place in low-inflammation reactions in patients with different CHF conditions. Therefore, we aimed to compare the cell count readings in HFrEF patients according to their condition (NYHA functional class) and to evaluate the correlations of cell counts with platelet aggregation, MPV, and inflammation markers (C-reactive protein and fibrinogen concentration) in chronic HFrEF. The following findings would allow a better understanding of the differences in chronic HFrEF conditions, could give a base for comprehensive research, and could contribute to the discovery of new possible treatment targets and the application of individual treatments according to patients’ conditions.

## 2. Material and Methods

### 2.1. Study Population

A total of 356 patients admitted to the Department of Cardiology of Kaunas Clinical Hospital of Lithuanian University of Health Sciences between 1 January 2016 and 1 March 2018 and diagnosed with CHF were screened for inclusion. The data for 185 patients diagnosed with chronic heart failure with reduced ejection fraction (systolic heart failure class I‒IV according to the New York Heart Association (NYHA)) who had not been taking any antiaggregants in the last two weeks and had experienced no other factors affecting platelets and had given written informed consent to participate were included in the study. The diagnosis of CHF was made following the guidelines for the diagnostics and treatment of heart failure approved by the European Society of Cardiology [14]. Patients with renal failure (eGFG < 90 mL/min.), acute and chronic infection, acute coronary syndromes, connective tissue or malignant disease, diabetes mellitus, and consuming platelet-affecting agents and smoking were excluded from the study (171). Table 1 shows the characteristics of patients.

### 2.2. Tests and Blood Sampling

The results of heart echoscopy (TEE) testing, fibrinogen and cortisol concentrations, complete blood counts, and a 6 min walking test were assessed and platelet aggregation was determined after the patients’ admission to the hospital. Blood samples for the complete blood counts and NT-pro BNP testing were taken from the forearm vein with a 20 G needle into 4.5 mL vacuum tubes with ethylendiamintetraacetic acid (EDTA) and for complete blood count testing were put into a COULTER LH 780 hematological analyzer (Brea, CA, USA). Blood for NT-pro BNP testing was centrifuged for plasma preparation and analyzed with a Tosoh AIA 2000 instrument (Tosoh Corporation, Tosoh, Japan). In order to investigate platelet aggregation, blood samples were taken from the forearm vein into 5 mL vacuum tubes with 3.8% sodium citrate. In order to prepare platelet-rich plasma, the blood was centrifuged at 100× *g* for 15 min at room temperature. Platelet-poor plasma was obtained when the rest of blood was centrifuged at 1000× *g* for 30 min. Platelet aggregation was investigated in platelet-rich plasma using the aggregometer (Chrono-Log, Havertown, PA, USA) with the standard Born method [15]. Adenosine diphosphate (ADP) and epinephrine (ADR) (at 3.8 and 4.5 mM (the final working concentration), using a Chrono-log P/N 384) were used for aggregation induction. Fibrinogen concentrations were determined in plasma with the Clauss method using a Bios-4 semiautomatic analyzer and Diagnostika Stago reagents [16,17]. Blood samples used to assess cortisol concentrations were taken in 4.5 mL vacuum tubes. Serum cortisol levels were measured twice a day (08:00–09:00 and 15:00–16:00) via automated enzyme-linked immunosorbent assay (ELISA) using the Roche Diagnostics ES700 instrument (Lewes, UK).

All investigations were approved and conducted in accordance with the guidelines of the local bioethics committee and adhered to the principles of the Declaration of Helsinki and Title 45, U.S. Code of Federal Regulations, Part 46, Protection of Human Subjects (revised 15 January 2009, effective 14 July 2009). The study was approved by the Regional Bioethics Committee at the Lithuanian University of Health Sciences (No. BE-2-102).

### 2.3. Statistical Analysis

For the statistical analysis, we used the Statistical Package for the Social Sciences (IBM SPSS 20) for Windows. Categorical variables were defined as percentages and comparisons were made using the chi-square test. Quantitative variables are expressed as means ± standard deviations, and for the comparison of variables between the groups Welch’s test (for the parameters showing the nonparametric distribution and to compare more than two groups) was used. The results were assessed using post hoc analysis. Correlation analyses were performed using a Pearson’s correlation test. Here, *p* values < 0.05 were accepted as statistically significant.

## 3. Results

### 3.1. Clinical Characteristics of the Patients

The study population consisted of 185 patients with CHF (with left ventricular ejection fraction (LVEF < 50%)): 26 NYHA I patients, 78 NYHA II patients, 54 NYHA III patients, and 27 pa NYHA IV patients. The baseline characteristics of the CHF patients are shown in the Table 1. The mean left ventricular ejection fraction (LVEF) value for NYHA I patients was bigger in comparison with NYHA III and IV (*p* < 0.015) patients. The 6 min walking test result decreased with the deterioration of patient state (*p* < 0.005). The mean BMI value for NYHA IV patients was lower than for NYHA I and II patients (*p* < 0.029). The SBP value was highest for NYHA I group patients (*p* < 0.015). The DBP value for NYHA IV patients was lower than in NYHA I group (*p* = 0.015). There were no differences in medications used by patients (Table 2). NT-proBNP concentrations gradually increased from NYHA I to NYHA IV groups.

### 3.2. The Differences in Cell Readings between CHF Patient Groups

A comparison of cell counts between the groups is shown in Table 3. There were no differences between platelet counts (PLT) and leucocyte counts. MPV values increased with deterioration of patient state (*p* < 0.005). The mean lymphocyte count and percentage were the lowest in NYHA IV group (*p* = 0.005 and *p* < 0.009, respectively). The neutrophil percentage in NYHA IV was higher than in NYHA III and I groups (*p* = 0.034). The neutrophil count in NYHA III group was lower than in NYHA IV (*p* = 0.028). The monocyte percentage and count in NYHA IV group was statistically significantly higher than in NYHA I group (*p* = 0.045 and *p* = 0.014, respectively).

### 3.3. The Differences in the Other Laboratory Readings between CHF Patient Groups

There were no statistically significant differences in CRP, fibrinogen, and morning cortisol (cortisol_m_) and evening cortisol (cortisol_e_) concentrations between the patient groups. The cortisol concentration difference between the morning and evening concentrations (cortisol_m-e_) was statistically significantly lowest in NYHA IV group in comparison with NYHA III and NYHA I groups (90.18 ± 162.09, 108.82 ± 161.89, and 121.04 ± 179.16, respectively, *p* < 0.021) (Table 4).

The result for ADP-induced platelet aggregation was significantly higher in NYHA III group compared to NYHA II and I groups (75.38 ± 8.94, 71.50 ± 11.26, and 68.33 ± 9.66, respectively, *p* < 0.023). The result for ADR-induced platelet aggregation was higher in NYHA III group than in NYHA I group (81.02 ± 11.85 and 72.88 ± 11.91, respectively, *p* < 0.002, Table 4).

### 3.4. The Correlation between Clinical and Complete Blood Count Readings

Data from the all patients were pooled to analyze the correlation between laboratory and clinical cell count readings. Patient age correlated with MPV (r = 0.214, *p* = 0.006), monocyte percentage (r = 0.293, *p* = 0.0001), and monocyte and lymphocyte counts (r = 0.180, *p* = 0.022 and r = 0.250, *p* = 0.001 respectively, Table 5). NYHA functional class correlated with MPV (r = 0.311, *p* = 0.0001), lymphocyte count (r = −0.186, *p* = 0.026), monocyte count (*p* = 0.172, *p* = 0.041), and monocyte percentage (r = 0.212, *p* = 0.011, Table 5). PLT correlated with leucocyte count (r = 0.306, *p* = 0.0001), neutrophil count (r = 0.183, *p* = 0.041) and monocyte count (r = 0.218, *p* = 0.014, Table 5). Monocyte count weakly correlated with CRP (r = 0.371, *p* = 0.0001) and with NYHA grouping (r = 0.172, *p* = 0.041). Monocyte percent correlated with NYHA (r = 0.212, *p* = 0.011). MPV correlated with monocyte percent (r = 0.419, *p* = 0.0001), and monocyte count (r = 0.317, *p* = 0.0001) (Table 5). Leucocyte count correlated with PLT (r = 0.306, *p* = 0.00001). BMI correlated with NYHA and lymphocyte count and percentage (r = −0.210, *p* = 0.007; r = 0.319, *p* = 0.029; and r = 0.257, *p* = 0.004, respectively; Table 5). CRP correlated with neutrophil count (r = 0.379, *p* = 0.0001) and monocyte count (r = 0.371, *p* = 0.0001), PLT (r = 0.307, *p* = 0.008) and conversely correlated with lymphocyte percent and PLT (r = −0.230, *p* = 0.021).

Despite the fact that there was no difference in age between the patient groups, age correlated with NYHA and MPV (Table 5). Therefore, a partial correlation assessment was run to determine the relationship between an individual’s MPV and NYHA whilst controlling for age. There was a weak partial correlation between MPV and NYHA whilst controlling for age, which was statistically significant (*r* = 0.277, *p* = 0.001). However, zero-order correlations showed that there was a statistically significant weak correlation between MPV and age (*r* = 0.295, *p* = 0.0001), indicating that age had very little influence in controlling for the relationship between MPV and NYHA functional class.

A partial correlation was run to determine the relationship between an individual’s MPV and monocyte and lymphocyte counts whilst controlling for age. There was a weak partial correlation between MPV and monocyte and lymphocyte counts whilst controlling for age, which was statistically significant (*r* = 0.175, *p* = 0.038 and *r* = −0.169, *p* = 0.044, respectively). Zero-order correlations showed that there was a statistically significant weak correlation between MPV and monocyte and lymphocyte counts (*r* = 0.212, *p* = 0.011 and *r* = −0.186, *p* = 0.044, respectively), indicating that age also had very little influence in controlling for the relationship between MPV and monocyte and lymphocyte counts.

### 3.5. Correlation between Laboratory and Clinical Readings

NYHA functional class correlated with patient age (r = 0.145, *p* = 0.049) and LVEF (r = −0.223, *p* = 0.002). Morning cortisol concentration correlated with platelet aggregation, induced with ADR (r = 0.238, *p* = 0.046). CRP correlated with NT-proBNP, fibrinogen, and cortisol_e_ concentrations (r = 0.203, *p* = 0.005, r = 0.408, *p* = 0.0001, r = 0.329, *p* = 0.001, respectively; Table 6).

### 3.6. Correlations between Complete Blood Counts and Other Laboratory Readings

PLT correlated with ADP (r = 0.313, *p* = 0.003), fibrinogen concentration (r = 0.180, *p* = 0.042), CRP (r = 0.307, *p* = 0.008, Table 6), and monocyte count (r = 0.218, *p* = 0.014, Table 5 and Table 6). MPV correlated with fibrinogen concentration (r = 0.244, *p* = 0.004, Table 6). Leucocyte count correlated with morning cortisol concentration (r = 0.238, *p* = 0.015) (Table 6). Neutrophil count correlated with PLT (r = 0.183, *p* = 0.041, Table 6), fibrinogen concentration (r = 0.308, *p* = 0.0001), evening cortisol concentration (r = 0.256, *p* = 0.009), and CRP (r = 0.378, *p* = 0.0001). Neutrophil percentage correlated with evening cortisol concentration (r = 0.264, *p* = 0.007). Lymphocyte percentage correlated with fibrinogen concentration (r = −0.174, *p* = 0.03), CRP (r = −0.220, *p* = 0.028), and evening cortisol concentration (r = −0.246, *p* = 0.012). Monocyte percentage correlated with fibrinogen concentration (r = 0.175, *p* = 0.03). Monocyte count correlated with morning cortisol, fibrinogen concentration (r = 0.279, *p* = 0.004 and r = 0.315, *p* = 0.0001, respectively), CRP (r = 0.371, *p* = 0.0001), and PLT (r = 0.218, *p* = 0.014; Table 5 and Table 6). Differences between the morning and evening cortisol concentrations correlated with neutrophil percentage (r = −0.244, *p* = 0.013), lymphocyte percentage (r = 0.256, *p* = 0.003), and lymphocyte count (r = 0.295, *p* = 0.003 and r = 0.295, *p* = 0.003, respectively; Table 6).

### 3.7. Correlations between Complete Blood Count Readings and Drug Usage

It was found that PLT, MPV, neutrophil and lymphocyte percentages, and neutrophil and lymphocyte counts weakly correlated with some drug usage (Table 7).

## 4. Discussion

Platelets are presented as cellular mediators of inflammation and atherogenesis via interactions with leukocytes (monocytes, lymphocytes, and neutrophils) and the endothelium [8]. A strong line of evidence has shown the close interrelation between inflammation and thrombosis [18]. A literature review showed that CHF patients have higher MPV and increased whole blood aggregation [18,19], and these readings have been found to be higher in patients in worse condition [20]. Our previous work showed MPV to correlate with age. Therefore, we aimed to assess whether platelet and complete blood readings differed in the chronic HFrEF groups according to NYHA grouping, in which patients did not differ by age, and to identify if components of the complete blood counts correlated with platelet readings.

### 4.1. Platelets

We found that mean MPV was the highest in the NYHA IV group. MPV increases with the deterioration of a patient’s state. Platelet aggregation induced by ADP and ADR was the highest in NYHA III group as compared to NYHA II and I groups. A weak correlation was found between MPV and fibrinogen concentration and a moderate correlation was found between MPV and NYHA grouping.

Some studies [21,22,23] have shown that smoking, hypertension, diabetes mellitus, dyslipidemia, and obesity are associated with increased MPV. We excluded smoking patients and diabetes mellitus patients from our study. BMI and SBP gradually decreased with increasing NYHA grouping in our studied HFrEF patients. Accordingly, DBP was statistically lower in NYHA IV in comparison with NYHA II and it had a tendency to decrease gradually. There was no correlation between lipid concentration and MPV. Therefore, the MPV increase in our study was not affected by hypertension, dyslipidemia, or obesity.

It was shown that MPV results depend on the time elapsed from the venipuncture and should be done within 120 min after venipuncture [24]. We investigated the complete blood counts in accordance with the requirements for the influence of elapsed time on MPV.

Increased levels of platelet activation markers in HFrEF patients in comparison with healthy controls have been reported in the literature [19,25]. The MPV has been previously shown to associate with VTE [26]. A higher MPV indicates that larger platelets on average have relatively larger contact surfaces and may be more reactive. Platelets with increased volumes could express higher quantities of adhesion molecules (such as P-selectin), which could increase the aggregation and adhesion. One study found higher platelet surface P-selectin levels in patients with acute HF (AHF) as compared to stable HF. This demonstrated an association between abnormal platelet activation and worsening cardiac function in HF [27,28]. Some studies have suggested that changes in hemodynamics and vascular system architecture facilitate blood stasis and predisposition to platelet aggregation [29]. Altered blood flow leads to an increase in circulating catecholamines and activation of the renin–angiotensin system, both of which contribute to platelet activation [30]. The MPV has been recognized as a strong indicator of platelet activation and independent risk factor for thromboembolic disease in acute myocardial infarction patients [31]. However, despite MPV gradually increasing in patient groups and correlating with NYHA grouping, platelet aggregation was found to increase only in the NYHA III group but not in NYHA IV group in our study. There was no correlation found between MPV and platelet aggregation either. The same increase in platelet aggregation in CHF groups was found in our previous work [20]. Therefore, the relationship between MPV and platelet activity in chronic HFrEF should be investigated by measuring additional platelet readings. MPV could serve as an additional marker of HF severity.

A moderate correlation between MPV and PLT (r = −0.41, *p* = 0.001) in patients with AHF has been stated in the literature [32]. We did not find such a correlation in chronic HFrEF patients. However, a tendency was seen for the PLT to decrease with the deterioration of NYHA.

Some authors have found a moderate correlation between MPV and BNP concentrations (r = 0.410; *p* = 0.001) in AHF patients [32]. We did not find such a correlation in HFrEF patients, however we found a moderate correlation between MPV and NYHA grouping. Therefore, our findings support a relationship between increased MPV levels and functional and hemodynamic parameters of disease severity.

It has been stated that heart failure (HF) patients have increased levels of soluble P-selectin and platelet-derived adhesion molecules [5], which indicate more active platelets. Additionally, sCD40 L was also shown to be increased in CHF patients when compared with healthy controls and to be correlated with NYHA and decreased ejection fraction [33]. The earlier described upregulation of CD40 and CD 154 on activated platelets [34] and the described expression of CD154 and P-selectin being positively correlated with increasing NYHA grouping and with the highest levels being found in NYHA IV patients (*p* < 0.001) [8] are in accordance with our findings that platelet aggregation increases with the deterioration of a patient’s state.

We failed to find any work about platelet aggregation differences in chronic HFrEF groups according to NYHA. Mostly, platelet activation markers were compared only between all CHF and control groups, but not between NYHA groups. Increased platelet activity in chronic, stable HFrEF was reported in one study, while CD62P was higher in NYHA III–IV in comparison with NYHA I–II (*p* = 0.036) [19]. The authors concluded that platelet activation may be related to HF. Another study presented the highest CD154 and P-selectin levels found in NYHA IV patients [8]. This was in consensus with our findings that MPV increased with the deterioration of a patient’s state and that the platelet aggregation was higher in the NYHA III group compared to NYHA II and I groups. However, it remains unclear whether the platelet aggregation increase in chronic HFrEF NYHA groups depended on the platelet specificity in NYHA VI or rather the sample size in this group was too small (*n* = 27).

A strong fibrinogen correlation with PLT was described in coronary heart disease patients [35]. Coagulation factor I, fibrinogen, is directly involved in the coagulation process and is associated with platelet aggregation. Accordingly, it is one of the acute-stage proteins and takes part in inflammatory reactions. We found a weak correlation between fibrinogen concentration and MPV and between PLT and CRP in chronic HFrEF patients. This may be explained by the documented role of interleukin 6 in hemostasis. There are some reports about the ability of all cells involved in the atherosclerotic process to produce acute-phase reaction cytokines, particularly IL-6, to cause increases in both acute phase protein and thrombopoietin plasma levels, suggesting the stimulation of PLT production [35].

Although uric acid measurement was not directly the aim of our work, it is necessary to mention that uric acid is stated as being related to inflammation and platelet activation and enhanced pro-thrombotic stimulus in acute coronary syndrome patients [36]. Some research studies have shown that measurement of uric acid level can improve the risk stratification in individuals at risk for CHF [37].

In conclusion, our results confirm a correlation between MPV and CHF patient condition and support a relationship between increased MPV levels and functional and hemodynamic parameters of disease severity. The correlations between fibrinogen concentration and MPV and between PLT and CRP support the low inflammation in chronic HFrEF patients and its relationship with platelets.

### 4.2. Monocytes and Neutrophils

Monocytes [8] and other leucocytes play important roles in chronic inflammatory processes [38]. Low total monocyte counts have been shown in several studies, which noted an association with the increased risk for mortality in HF patients [39,40,41]. However, more studies have supported an association of increasing monocyte count with worse outcomes [42,43]. It is known that outcomes depend on patient condition. Therefore, we examined the blood cell count differences in NYHA groups and the correlations with NYHA, LVEF, CRP, NT pro-BNP, and platelet readings.

Humans have three types of monocytes: Mon1 (85%), Mon2 (6%), and Mon3 (9%). Mon1 has been shown to express cytokines, whereas Mon 2 produces anti-inflammatory IL-10. Mon1 numbers are increased during HF decompensation [8,38]. Mon3 produces pro-inflammatory cytokines, such as interleukin 1 [44]. Several studies have highlighted the presence of increased levels of Mon 2 in human inflammatory diseases [45] and a correlation with NYHA, LVEF, and NT-proBNP [44,46], however another found no changes versus healthy control [47]. Mon 3 subset levels in CHF increased depending on how advanced NYHA was, on the worsening of LVEF, and on pro-BNP values (r = 0.534, *p* = 0.047) [44]. Patients with HFpEF have been found to have raised monocyte counts in comparison with healthy controls [48]. We found that in HFrEF patients, the mean monocyte count was the largest in the NYHA IV group. These findings are in accordance with most previous studies, where larger monocyte counts were associated with poor patient condition [42,43,49].

A reason why different studies present conflicting results may be because of different subsets of monocytes being predominant in CHF groups [39]. One work showed that Mon 3 gradually increased in NYHA groups (*p* = 0.0002) and negatively correlated with LVEF (r = −0.628, *p* = 0.022) [44]. We found an unexpected correlation between the total monocyte count and percentage and NYHA grouping. This supports the findings that monocyte count increases depending on chronic HFrEF disease severity.

Additionally, we found that monocyte count weakly correlated with PLT and moderately correlated with MPV and CRP. These correlations can be assumed as a relationship between inflammation and platelet production. In Charach’s work [39], it was stated that Hs-CRP did not differ in groups according to monocyte distribution in CHF patients. However, we found a moderate correlation between the monocyte count and CRP. It can be assumed that monocytes expand in response to this inflammatory environment. In pathological conditions, there is an override of uncontrolled inflammation, which leads to the exaggerated release of macrophages. and instead of tissue healing causes tissue damage with adverse remodeling. Therefore, trying to regulate the monocyte–macrophage balance could be a logical therapeutic strategy. In conclusion, the monocyte count correlation with CRP and the highest level being found in the NYHA IV group revealed that a monocyte count increase reflects the worsening of a patient’s condition and that this finding supports the inflammation present in chronic HFrEF patients.

The neutrophil count serves as an indicator for inflammation [50]. Neutrophils have been stated to be positively associated with heart failure and stroke incidence [51]. Nevertheless, neutrophils may also control to the resolution of inflammation and the return to tissue functionality [52].

High levels of CRP and of neutrophils have already been described in CHF patients. Levels of myeloperoxidase, which is a product of activated neutrophils, are excessive in CHF patients. It has been reported that myeloperoxidase levels were linked to functional indices of worsening heart failure [44], similarly to what we observed for monocytes in our work.

We found that the mean neutrophil count and percentage were the greatest in NYHA IV group and moderately correlated with fibrinogen concentration and CRP. Some authors have found that in heart failure patients, the incidence was greater among people with higher neutrophil counts, and that the greater the neutrophil count was, the bigger the hazard ratio [53]. Therefore, both neutrophils and monocytes may act as participants in the pro-inflammatory state of chronic HFrEF. If the neutrophil and monocyte counts for each patient can be regarded as indicators of inflammation, then neutrophil and monocyte counts would provide us with a marker that supplies prognostic information for CHF.

### 4.3. Lymphocyte

Low lymphocyte count or low percentage of lymphocytes has been associated with patient mortality in CHF [54].

In our work, the mean lymphocyte count was the lowest in the NYHA IV group and negatively correlated with NYHA grouping. Lymphocyte percentage negatively correlated with CRP, fibrinogen concentration, and evening cortisol concentration. Other authors have found that circulating lymphocyte subset Treg negatively correlated with BNP, LV chamber remodeling, and CRP [55], indicating that the reduction of Treg-mediated immunosuppression may promote worsening of these parameters. The finding that decreases Treg is associated with reduced LV systolic function [55] supplemented our finding that the mean lymphocyte count was the lowest in NYHA IV group and negatively correlated with NYHA. In the review, it was seen that lymphopenia is more common in stressful conditions such as HF, due to the activation of the hypothalamic–pituitary–adrenal axis. The activation of this axis leads to cortisol secretion. The increased cortisol results in a decrease of lymphocytes [12]. Our results showed that the statistically significantly lowest Cortisol_m-e_ value was in NYHA IV. Evening cortisol concentration inversely correlated with lymphocyte percent. Cortisol_m-e_ correlated with the lymphocyte count and percentage. All of these results support the knowledge about conditions of greater stress in chronic HFrEF and its influence on lymphocyte counts.

We found that CRP correlated with NT-proBNP. This finding suggests an association between neurohumoral activation and inflammation in the setting of HFrEF. In conclusion, the lowest lymphocyte count in NYHA IV and its negative correlations with NYHA grouping and evening cortisol concentration indicate a relationship between the disease severity and neurohormonal parameters, which had an influence on lymphocyte count. Therefore, lymphocyte count could be used as an additional marker for evaluation of a patient’s condition.

## 5. Study Limitations

Our study had some limitations. One of them was the small sample size. This might be why we did not observe a correlation between CRP and LVEF and NYHA functional class, or between MPV and PLT, which are described in the literature. The other limitation is the absence of healthy subjects or HFrEF groups that could help in defining pathological links.

We did not measure the levels of inflammatory markers, such as TNF-α, IL-6, and others. Therefore, we could not compare those inflammatory factors. Additionally, we did not measure other platelet activation readings, meaning we can evaluate platelet activity only when we have platelet aggregation results.

The range of CRP levels at enrollment was wide, and it was unable to adjust for confounding factors such as subclinical infection. Interpretation of the present results is limited because monocytes and lymphocyte phenotypes were not investigated. Different monocyte and lymphocyte subsets have distinct mechanisms. Therefore, their differences in different patient condition groups and relationships with platelet and inflammation markers remain to be investigated. However, the correlations we found may serve as reliable markers in the additional evaluation of chronic HFrEF severity.

The main strength of our study was the broad age range of the population, without smoking or diabetes mellitus influencing platelet readings. The influence of medications on blood count and platelet parameters was evaluated too. Blood samples were collected at baseline and measured with standardized tests, allowing comparability with other studies. Standardization is particularly important for MPV measurement.

In addition, this was an observational and nonrandomized study; it can be accepted as a prototype of further prospective and randomized studies to compare the effects of the blood cell counts on platelet and inflammatory readings in chronic HFrEF.

To the best of our knowledge, this is the first work where blood cell count and platelet readings of MPV are compared between chronic HFrEF patient groups according to NYHA grouping, and in which the correlations between platelet, blood cell count, CRP, NT-proBNP, and cortisol readings were evaluated.

## 6. Conclusions

We expanded understanding of differences between chronic HFrEF patients groups according to NYHA grouping. Our work revealed differences of platelet readings; platelet aggregation and MPV values; and blood cell monocyte, lymphocyte, and neutrophil counts in those groups, as well as correlations between platelet and inflammation readings. According to these findings we can state that: (1) MPV could be assumed as an additional reading reflecting a patient’s condition and that the idea that MPV values help in identifying patients at risk of hypercoagulable state should be evaluated in more extensive studies; (2) neutrophil and monocyte increases in HFrEF could indicate a higher inflammatory state in chronic HFrEF.

## Figures and Tables

**Table 1 medicina-57-00176-t001:** Clinical characteristics of patients according to the New York Heart Association (NYHA) class.

Clinical Variables	I NYHA	II NYHA	III NYHA	IV NYHA	*p* Value
	(*n* = 26)	(*n* = 78)	(*n* = 54)	(*n* = 27)	
Age, years (mean ± SD)	51.50 ± 7.99	53.95 ± 13.53	54.37 ± 9.49	57.78 ± 14.86	0.295
Male (*n* (%))	21 (80.8)	67 (85.7)	48 (88.9)	18 (66.7)	0.070 *
Female (*n* (%))	5 (19.2)	11 (14.1)	6 (11.1)	9 (33.3)	0.070 *
Left ventricular ejection	34.28 ± 11.45	30.80 ± 10.99 ^d,e^	27.71 ± 12.25	25.76 ± 11.07	<0.015
fraction (%, mean ± SD)					
6 min. walking test, m (mean ± SD)	518.65 ± 107.74 ^b,c^	471.23 ± 82.96 ^d,e^	378.9 ± 127.10	273.18 ± 143.18	<0.005
BMI (median, min-max)	29.38 (21–37) ^a,c^	27.38(19–43)	26.54(21–44)	25.00(21–45)	<0.029
SBP (mean ± SD)	138.38 ± 18.39 ^abc^	124.81 ± 17.84	123.85 ± 20.73	115.56 ± 23.99	<0.015
DBP (mean ± SD)	85.27 ± 10.83 ^c^	82.12 ± 12.45	81.13 ± 14.33	75.30 ± 11.95	0.015

BMI—body mass index; SBP—systolic blood pressure; DBP—diastolic blood pressure; LVEF—left ventricular ejection fraction; a—statistically significant difference between NYHA I and NYHA II patients; b—statistically significant difference between NYHA I and NYHA III patients; c—statistically significant difference between NYHA I and NYHA IV patients; d—statistically significant difference between NYHA II and NYHA III patients; e—statistically significant difference between NYHA II and NYHA IV patients; *—the comparisons were made using the chi-square test; Welch’s test was used for other variables.

**Table 2 medicina-57-00176-t002:** Chronic disease medications and outcomes.

Variables	I NYHA (*n* = 26)	II NYHA (*n* = 78)	III NYHA (*n* = 54)	IV NYHA (*n* = 27)	*p*-Value
Diuretics (*n*)	9 (28%)	29 (37%)	31 (50%)	11 (35%)	0.28
Beta-blockers (*n*)	18 (56%)	48 (62%)	35 (56%)	12 (38%)	0.211
ACE-inhibitors (*n*)	15 (57.7%)	43 (55.1%)	29 (53.7%)	14 (51.9%)	0.976
Nitrates (*n*)	2 (7%)	3 (4%)	5 (9.3%)	3 (11.5%)	0.534
Digoxin (*n*)	1 (3.8%)	5 (6%)	4 (7%)	6 (22%)	0.12
Statines (*n*)	3 (11.5%)	3 (4%)	4 (7%)	1 (4%)	0.91
Heparine (*n*)	0	1 (1%)	4 (7%)	1 (4%)	0.201
Calcium channel blockers (*n*)	0	1 (1%)	1 (2%)	3 (10%)	0.051
Thrombosis (*n*)	0	3 (15%)	0	2 (12.5%)	0.413

The comparisons were made using the chi-square test.

**Table 3 medicina-57-00176-t003:** Complete blood count readings for the CHF patients according to NYHA groups.

Readings	I NYHA (*n* = 26)	II NYHA (*n* = 78)	III NYHA (*n* = 54)	IV NYHA (*n* = 27)	*p* Value
Platelet count × 10^9^/L (mean ± SD)	227.29 ± 54.52	222.83 ± 63.09	217.83 ± 55.29	218.95 ± 124.89	0.972
MPV, Fl (mean ± SD)	9.25 ± 0.46 ^a^	9.65 ± 1.22 ^b^	9.78 ± 1.15 ^c^	10.75 ± 1.2	<0.005
Leukocyte count × 10^9^/L (mean ± SD)	7.17 ± 2.11	7.10 ± 2.07	6.82 ± 1.83	7.31 ± 1.85	0.878
NEU, % (mean ± SD)	58.65 ± 9.55 ^a^	62.61 ± 8.02	58.56 ± 10.35 ^c^	65.86 ± 8.99	0.034
NEU count × 10^9^/L (mean ± SD)	4.20 ± 1.32	4.52 ± 1.34	4.11 ± 1.64 ^c^	4.99 ± 1.73	0.028
LIMPHO, % (mean ± SD)	28.53 ± 8.82 ^a^	24.48 ± 7.14	28.05 ± 9.66 ^c^	20.35 ± 6.15	<0.009
LIMPHO count × 10^9^/L (mean ± SD)	2.09 ± 0.85 ^a^	1.70 ± 0.56	1.88 ± 0.59 ^c^	1.44 ± 0.40	0.005
MONO, % (mean ± SD)	8.23 ± 1.84 ^a^	8.7 ± 2.77	8.90 ± 3.41	10.67 ± 3.63	0.045
MONO count × 10^9^/L (mean ± SD)	0.59 ± 0.21 ^a^	0.61 ± 0.25	0.60 ± 0.25	0.76 ± 0.27	0.014

ANOVA was used for comparisons. Note: a—statistically significant difference between NYHA I and NYHA IV patients; b—statistically significant difference between NYHA II and NYHA IV; c—statistically significant difference between NYHA IV and NYHA III. MPV—mean platelet volume; NEU—neutrophil; LIMPHO—lymphocyte; MONO—monocyte.

**Table 4 medicina-57-00176-t004:** NT-proBNP and CRP readings for the CHF patients according to NYHA groups.

Readings	I NYHA (*n* = 26)	II NYHA (*n* = 78)	III NYHA (*n* = 54)	IV NYHA (*n* = 27)	*p* Value
NT-proBNP, ng/L (median, min-max)	222.9 (29.0–2160) ^a,b,c^	702 (45.2–6501)	1496.5 (159–15377)	2095.5 (450.9–7649)	<0.007 *
CRP, mg/L (median, min-max)	3.12 (1.0–15.50)	3.13 (0.60–40.80)	3.15 (1.00–41.00)	4.62 (2.20–90.50)	0.668 *
Fibrinogen, g/L (median, min-max)	3.59 (2.6–5.3)	3.60 (2.2–7.9)	3.74 (2.3–7.5)	4.00 (2.7–5.1)	0.395 *
Cortisol_m_, nM (mean ± sd)	470.35 ± 165.71	476.32 ± 157.17	480.01 ± 166.90	439.14 ± 160.10	0.995
Cortisol_e_, nM (mean ± sd)	349.31 ± 154.37	369.86 ± 122.54	370.18 ± 119.49	440.87 ± 143.39	0.059
Cortisol_m_-cortisol_e_, nM (mean ± sd)	121.04 ± 179.16 ^c,d^	78.14 ± 158.19	108.82 ± 161.89	90.18 ± 162.09	<0.021
Platelet aggregation, ADP,% (mean ± sd)	68.33 ± 9.66 ^b^	71.50 ± 11.26 ^f^	75.38 ± 8.94	70.62 ± 12.11	<0.023
Platelet aggregation, ADR,% ( mean± sd)	72.88 ± 11.91 ^a,b^	80.94 ± 9.65	81.02 ± 11.85	78.42 ± 15.46	0.002

Note: *—The comparisons were made using the Welch’s test. ANOVA was used for other comparisons. Note: a—statistically significant difference between NYHA I and NYHA II patients; b—statistically significant difference between NYHA I and NYHA III patients; c—statistically significant difference between NYHA I and NYHA IV patients; d—statistically significant difference between NYHA III and NYHA IV patients; f—statistically significant difference between NYHA II and NYHA III patients; MPV—mean platelet volume; CRP—C reactive protein; Cortisol_m_—morning cortisol concentration; Cortisol_e_—evening cortisol concentration; Cortisol_m-e_—difference between morning and evening cortisol concentrations; ADP—adenozinediphosphate; ADR—adrenaline; NT-proBNP—N-terminal pro b-type natriuretic peptide.

**Table 5 medicina-57-00176-t005:** Correlations between complete blood count and clinical readings.

Readings	Age	BMI	NYHA	PLT	MPV	CRP
Leucocyte count × 10^9^				0.306, 0.00001		
Neutrophil count × 10^9^				0.183, 0.041		0.379, 0.0001
Lymphocyte count × 10^9^	0.250, 0.001	0.319, 0.029	-0.186, 0.026			
Lymphocyte%		0.257, 0.004				−0.230, 0.021
Monocyte count × 10^9^	0.180, 0.022		0.172, 0.041	0.218, 0.014	0.317, 0.0001	0.371, 0.0001
Monocyte%	0.293, 0.0001		0.212, 0.011		0.419, 0.0001	
MPV	0.214, 0.006		0.311, 0.0001			
PLT						0.307, 0.008
Age			0.145, 0.049		0.250, 0.0001	
NYHA	0.145, 0.049	−0.210, 0.007			0.311, 0.0001	

MPV—mean platelet volume; CRP—C reactive protein; BMI—body mass index; PLT—platelet count.

**Table 6 medicina-57-00176-t006:** Correlations between complete blood count, clinical and laboratory readings.

Readings	CRP	Fibrinogen Concentration	Cortizol_m_	Cortizol_e_	Cortizol_m-e_	ADP
PLT	0.307, 0.008	0.180, 0.042				0.313, 0.003
NT-proBNP	0.203, 0.005	0.306, 0.0001	0.238, 0.015			
Neutrophil%				0.264, 0.007	−0.244, 0.013	
Neutrophil count × 10^9^	0.378, 0.0001	0.308, 0.0001		0.256, 0.009		
Lymphocyte%	−0.220, 0.028	−0.174, 0.03		−0.246, 0.012	0.256, 0.009	
Cortisol_e_		0.329, 0.001			0.295, 0.003	
Monocyte count × 10^9^	0.371, 0.0001	0.315, 0.0001	0.279, 0.004			
MPV		0.244, 0.004				

PLT—platelet count; CRP—C reactive protein; Cortisol_m_—morning cortisol concentration; Cortisol_e_—evening cortizol concentration; Cortisol_m-e_—difference between morning and evening cortisol concentrations; ADP—adenozinediphosphate; NT-proBNP—N-terminal pro b-type natriuretic peptide.

**Table 7 medicina-57-00176-t007:** Correlations between complete blood count readings and drug usage.

Readings	PLT	MPV	Neutro-Phyle%	Neutro-Phyle Count
Diuretics (r, *p*)	−1.69, 0.041	−1.66, 0.034	−0.187, 0.017	
Nitrates (r, *p*)			0.172, 0.029	0.196, 0.013
Statines (r, *p*)	0.211, 0.011		−0.166, 0.034	
Calcium channel blockers (r, *p*)		0.269, 0.0001		
Digoxin (r, *p*)		0.175, 0.025		
Heparine (r, *p*)		0.216, 0.005		

PLT—platelet count; MPV—mean platelet volume. Stepwise regression analyses used with drug usage as predictors or determinants and measured readings showed that diuretics, calcium channel blockers, and digoxin were weak independent predictors for MPV (R^2^ = 0.03, *p* < 0.027; R^2^ = 0.07, *p* < 0.001; R^2^ = 0.03, *p* < 0.027, respectively). Diuretics and statins were weak independent predictors for PLT (R^2^ = 0.029, *p* < 0.041 and R^2^ = 0.045, *p* < 0.011, respectively). Diuretics, nitrates, and statins were weak independent predictors for neutrophil percentage (R^2^ = 0.035, *p* < 0.017, R^2^ = 0.029, *p* < 0.029, R^2^ = 0.028, *p* < 0.034, respectively).

## Data Availability

The study did not report any data.

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
