# Peer review of "Platelet Activity and Its Correlation with Inflammation and Cell Count Readings in Chronic Heart Failure Patients with Reduced Ejection Fraction"

_medicina, 2021, doi:10.3390/medicina57020176_

Round 1
Reviewer 1 Report
Proposed paper is interesting and well written. However, some revision are needed before it can be accepted for pubblication:
- The main limitation of the paper is the absence of a control group of healthy subjects and/or a group of HFpEF that could help defining pathological link. Could it be added to the paper? if not this should be listed as the principal limitation of the paper.
- Another important limitation is the small number of subjects in the most important subgroup (NYHA class I and IV). Also this should be encountered as an important limitation in the relative section.
- Correlation of different inflammatory markers one among other is not so interesting and is expected. It should be removed from the abastract since it is not informative and also from the text. More interesting correlatio suh as the evaluated markers with NYHA class or EF values that are presented but should be higlilghtened.
- Too much tables and too much correlations are presented. Tables should be added (example 3 and 4 should constitute a unique table) and correlation to be showed should be selected from the authors and only the most imformative one presented.
- What about sacubitril-valsartan in the therapies? at the time of last patients enrolled it was present on the market, are some patients taking it?
- Something very new in the field of heart failure is it's correlation with uric acid. Although this is not directly the aim of the current paper uric acid is surely related to inflammation and also platelet activation. So, this need to be cited in the discussion toghether with some recent findings on this topic (i.e. J Hypertens. 2021 Jan;39(1):62-69. and Eur J Intern Med. 2020 Dec;82:62-67.).
Author Response
Thanks to the Reviewer for carefully reviewing the article and helpful comments. We have corrected the article in the light of the comments. It improved the quality of the article. The following corrections have been made:
- We do not have the results of healthy subjects or a group of HFpEF, which could help defining pathological link. So we had listed it in the chapter of limitations.
- We emphasized the importance of a small number of subjects in the section Study limitations.
- Correlations of different inflammatory markers one among other has been removed from the abstract and from the text (in 2.4 and 2.6 chapters, and in 6-7 tables). Correlations between monocyte count and NYHA has been highlighted in part “Monocytes and neutrophils “ in the chapter Discussion.
- The number of tables hve been reduced: the tables 6 and 7 have been merged to one. The most informative correlations have been selected and presented in the new table.
- Patients that were included in the study were not prescribed sacubitril-valsartan. It is possible that patients were not prescribed this medicine because it was not on the list of reimbursable medicines yet.
- The knowledge about uric acid investigations are included in the chapter Discussion and the proposed articles are cited. The absence of uric acid measurements in our work is involved in chapter Study limitations.
- Conclusions have been revised and shortened.

Reviewer 2 Report
Dear Authors,
firstly I would like to thank you for submitting your paper. The topic is quite interesting and always current.
The idea of comparing blood cell count and MPV and the correlations with platelet activity, CRP, NT-proBNP and cortisol in heart failure patients divided according to NYHA class, could lead, following larger and more structured studies, to identify new markers of progression of the pathology and activation of the inflammatory cascade in this subset of patients and new therapeutic targets.
The study has several limitations (correctly identified and reported in the text in the appropriate section), first of all the sample size and the lack of some measurements of inflammatory indices, platelet activity markers and subanalyses (e.g. lymphocytic and monocytic phenotypes) that would have supported the theses expressed in the conclusions. The results obtained through a precise statistical analysis, however, the numerous correlations analyzed, the argued discussion presented supported by an in-depth study of the literature and the numerous insights provided, make this manuscript a starting point for future prospective studies.
I have some suggestions:
- although the article was written with a correct use of the English language, there are several typos to be corrected (both in the text and in the tables - e.g. "cortizol") and there is the need to check the timeline of some sentences.
- the results and the discussion appear verbose, sometimes repetitive and unclear. It seems appropriate to me a profound revision of these sections, with a restructuring and better use of the tables to support the text.
Author Response
Thanks to the Reviewer for carefully reviewing the article and helpful comments. We have corrected the article in the light of the comments. It improved the quality of the article. The following corrections had been made:
1 “Cortizol” has been corrected into “cortisol” in the tables. Translator of our institution has revised the paper.
- The chapters Results and Discussion has been revised. Following corrections have been made:
- a) Correlations of different inflammatory markers one among other has been removed from the abstract and from the text (in 2.4 and 2.6 chapters, and in 6-7 tables). Correlations between monocyte count and NYHA (as more interesting) have been highlighted in part “Monocytes and neutrophils“ in the chapter Discussion.
- b) The number of tables have been reduced: the tables 6 and 7 have been merged to one. Only the most informative correlations have been selected and presented in the new table.
- c) Conclusions have been revised and shortened.

Round 2
Reviewer 1 Report
Authors replies to all the query raised and paper improves and can now be accepted for pubblication
Reviewer 2 Report
Dear Authors,
thanks for editing the paper as suggested.
The changes made to the abstract, tables and conclusions streamlined the manuscript. I still think that the results and especially the discussion section would have benefited from a revision in order to reduce repetitions and to make the messages reported more direct.
Overall, however, the manuscript now appears more readable and easier to understand.